# Stress Appraisals and Coping across and within Academic, Parent, and Peer Stressors: The Roles of Adolescents' Emotional Problems, Coping Flexibility, and Age

Melanie J. Zimmer-Gembeck [1,*] and Ellen A. Skinner [2]

1   School of Applied Psychology, Gold Coast Campus, Griffith University, Southport 4222, Australia
2   Department of Psychology, Portland State University, Portland, OR 97207, USA; skinnere@pdx.edu
*   Correspondence: m.zimmer-gembeck@griffith.edu.au; Tel.: +61-756789085

**Abstract:** The aim of this study was to determine whether adolescents' emotional problems, coping flexibility, age, and stress appraisals account for ways of coping, which include engagement and disengagement coping, with academic-, parent-, and peer-related stressful events. Stress appraisals were defined as perceived threats to the psychological needs of relatedness, competence, and autonomy. Models were fit at a higher order level, indicated by adolescents' appraisals and intended ways of coping with stress in three domains (i.e., academic, parent, and peer) and tested at the lower level within each domain. Adolescents ($N$ = 410; age 10–15; Mage = 12.5; 50% girls) reported their emotional problems (combined depressive and anxiety symptoms) and coping flexibility six months prior to completing an analogue task. The task involved viewing six short film clips portraying stressful events (e.g., obtaining a worse than expected exam grade or arguing with a parent) and reporting three stress appraisals and eight ways of coping after each stressor. The ways of coping were analyzed as four composite scores reflecting engagement coping (active coping, self-reliance) or disengagement coping (withdrawal coping, helplessness). In structural equation models, adolescents who appraised more threat reported more withdrawal coping and helplessness but also more active coping and self-reliance. Adolescents with more emotional problems appraised more threat and anticipated using less constructive ways of coping, whereas adolescents higher in coping flexibility intended to use more constructive ways of coping, with these associations sufficiently modeled at the general (across stress domains) level. Improvement in the model fit was found when appraised threat–coping associations were modeled at the lower (specific stressor domain) level, suggesting differences by stressor domain. Age was associated with more self-reliance and helplessness, with self-reliance being specific to parent stressors and helplessness specific to peer stressors.

**Keywords:** stress and coping; depression; social anxiety; psychological needs; adolescence



## 1. Introduction

The high rates of mental health problems among youth [1] are thought to be partly due to experiences of stressful life events and significant daily hassles [2]. Yet, when stressors are tackled successfully, they can also be foundational for positive development, such as developing a better understanding of the most constructive ways of coping with different types of stressors [3]. Therefore, experiences with stressors and coping are implicated in the development of emotional problems, but they are also important for developing coping beliefs and skills, such as an understanding of one's own ability to cope flexibly and competently with stressful events (coping flexibility). In turn, emotional problems and coping flexibility can have feedforward effects, impacting on future stress appraisals and ways of coping [4,5].

The above ideas follow directly from transactional conceptualizations of stress and coping [6,7] and developmental coping conceptualizations in particular [2,3,8]. These theories highlight the roles of stress appraisals and ways of coping as founded in personal

resources and vulnerabilities (such as emotional problems and coping flexibility). Stress appraisals include individual perceptions of the reasons, significance, and meaning of encounters with threatening or challenging events within contexts. Meaning comprises the information that organizes coping actions, with coping defined as "constantly changing cognitive and behavioral efforts to manage specific external and/or internal demands that are appraised as taxing or exceeding the resources of the person" [9] (p. 141). The specific ways of coping can vary from active or engagement coping to avoidant, withdrawal, or disengagement coping [10].

Throughout adolescence, young people face many situations that are threatening and distressing, providing numerous opportunities for learning about stress and ways to cope. Thus, through these experiences, all adolescents have had an opportunity to develop some sense of their own coping flexibility and capacity. However, specific types of stressors encountered by adolescents, which adolescents appraise as more threatening, and emotional problems and coping flexibility vary widely, thereby resulting in differences between adolescents and between different types of stressors. Moreover, age differences may still be found in how adolescents appraise and cope with stressors. Our aim in the present study was to consider whether adolescents' emotional problems, coping flexibility, and age uniquely relate to their stress appraisals and ways of coping. We considered ways of coping within each of three highly salient stressor domains for adolescents— academics, interactions with parents, and peer relationships [10–13]—and in general (across the three domains).

## 1.1. The Motivational Theory of Coping

The motivational theory of coping (MTC) was drawn upon for this study to identify the stress appraisals and ways of coping that should be founded in adolescents' emotional problems, coping flexibility, and age. The MTC proposed a developmental conceptualization of coping that draws from basic psychological need theory [14] and self-determination theory (SDT) [15] to identify stress appraisals, especially highlighting the important role of appraisals of threat to the psychological needs of relatedness, competence, and autonomy. These stress appraisals were expected to be shaped by adolescents' pre-existing vulnerabilities, resources, and experiences (e.g., emotional problems, coping flexibility, and age), but stress appraisals were also expected to be associated with ways of coping with stressors, which were consistent with transactional conceptualizations of stress and coping [6,7].

### 1.1.1. Stress Appraisals as Perceived Threats to Psychological Needs

In the MTC, environmental events can be called "stressors" (having potential for an impact on ill health and reduced motivation and engagement) when they are appraised as threatening basic psychological needs of relatedness, competence, and autonomy identified in the SDT. A threat to relatedness reflects an appraisal of an event as interfering with the psychological needs of connection, love, belongingness, and value to others [16,17]. An appraised threat to competence reflects interference in meeting the need to be effective in the environment and to understanding means–end relationships [2,18,19]. An appraised threat to autonomy reflects interference in meeting the needs of volition and agency without undue coercion by others [14,15,20].

Substantial research has identified the psychological needs of relatedness, competence, and autonomy as core to human behavior and social functioning. In addition, when individuals report higher need threat or frustration, they also have poorer mental and physical health (for a discussion of relatedness, see [16]; for a discussion of competence, see [19]; for a discussion of autonomy, see [15,19–21]). The MTC [8] built on these ideas to propose that individuals' perceived threats to relatedness, competence, and autonomy are part of stress appraisal processes. Such appraisals would also be related to ways of coping with stressful events whereby appraising more threat should interfere with constructive ways of coping. This interference would include undermining the most adaptive and useful ways of coping and promoting coping that may not repair personal distress or remedy

stressful events themselves. Thus, appraising more threat to relatedness, competence, and autonomy from stressful events should be associated with less engagement coping and more disengagement coping.

### 1.1.2. Ways of Coping

Ways of coping were also derived from the MTC [8] as well as research that reviewed and consolidated the hundreds of ways of coping that have been studied [22]. Ways of coping included four ways of engagement coping and four ways of disengagement coping that are common among adolescents [22,23]. Ways of engagement coping included problem-solving, support-seeking, information-seeking, and self-reliance. In the MTC, these ways of coping are more likely to repair threats to relatedness through protecting and approaching the available social resources; reduce threats to competence through finding solutions or more information to adjust actions to be more effective and to identify contingencies; and reduce the threat to autonomy by having more agency, providing opportunities to find new options, or flexibly adjusting preferences based on the available information and options. The ways of disengagement coping include social withdrawal, escape, giving up, and helplessness. These ways of coping are less likely to repair social problems, reduce opportunities for understanding contingencies, and may only result in more constraints on available choices. Notably, the MTC also identifies how heightened stress appraisals of threat can interfere with engagement coping and make disengagement coping more likely, suggesting that threat appraisals should be negatively associated with engagement coping and positively associated with disengagement coping.

### 1.2. Emotional Problems, Coping Flexibility, Stress Appraisals, and Ways of Coping

There has been a long history of examining how environmental and psychological stress and ways of coping relate to emotional problems, including an extensive body of research on internalizing behavior (e.g., depressive and anxiety symptoms or disorders) [2,5,8,11,24,25]. In general, there is support for concluding that emotional problems develop from stress experiences and prompt stressors into the future among adolescents [25–27]. Emotional problems have been related to a heightened sensitivity to stressful events and elevated threat responses [28–30]. Moreover, adolescents with more emotional problems have been found to use more maladaptive and fewer adaptive ways of coping (e.g., see [2,5,10,31]).

Not only are emotional problems relevant to stress appraisals and coping, but emerging views of human flexibility in many areas [7,32–36] have identified the positive downstream effects of coping flexibility on stress appraisals and ways of coping [32,37]. Adolescents who believe they are better able to cope with stress tend to cope more successfully with stressful events and to receive more interpersonal support [38,39]. Moreover, emotional problems and the belief in the capacity to cope are interrelated; adolescents who report more depressive and anxiety symptoms also report a poorer capacity to cope with stress [39].

One way of defining perceived coping capacity has been to capture the self-perception of having access to multiple ways of coping when needed. This approach comes from one core area of past research on coping flexibility [32,36,40], which has been defined as "individuals' subjective appraisals of their own ability to deploy diverse coping strategies to deal with environmental changes" [32] (p. 1585). Evidence suggests that coping flexibility, when measured in university students or adults, can mitigate the appraisal of threat when stressors occur [41,42] and also undergird more positive ways of coping and yield better wellbeing despite stressful events [37,43–45]. Although it has been rare to study adolescents' emotional problems and coping flexibility in a single study, one series of studies found that youth's coping flexibility was associated with less emotion dysregulation, more adaptive ways of coping with stressors such as cognitive reappraisals, and less rumination as a response to stressful events [46]. In other research, coping efficacy, which also captures beliefs in the capacity to cope with stress, was related to more adaptive ways of coping and fewer emotional problems [39].

### 1.3. Age, Stress Appraisals, and Ways of Coping

Adolescents' age may also relate to emotional problems, coping flexibility, stress appraisals, and ways of coping. Emotional problems, including both depressive and anxiety symptoms, are known to increase with age [47], and previous research has found that coping flexibility has a small positive association with age [48]. In addition, ways of coping show age-related patterns of change during adolescence. For example, self-reliance increases in middle adolescence, while support and information seeking might stay stable (although the source of support and information seeking might shift from parents to peers or become more specific to the stressor (for a review see [2]). Thus, given that adolescents in this study ranged from age 10 to 15, it was important to account for potential age effects on all study measures.

### 1.4. The Current Study

In summary, theory suggests how ways of coping with stress are grounded in stress appraisals, which we measured as the psychological threat attached to experienced stressful events. Very few previous studies have measured stress appraisals as appraised threats to relatedness, competence, and autonomy. Therefore, although hypotheses can be made based on theory [8], it is not known how these stress appraisals relate to ways of coping with stressful events, and no study has examined such relationships among adolescents, a time when stressors can be novel and especially threatening, coping skills are under development, and beliefs about coping abilities may be especially modifiable. To address these gaps in the research literature, we investigated whether adolescents' emotional problems (measured as depressive and social anxiety symptoms), coping flexibility (i.e., beliefs about the capacity to cope with stress), and age were predictive of stress appraisals and ways of coping and how stress appraisals related to adolescents' ways of coping with academic, parent, and peer stressors.

We tested models to quantify (1) the interrelationships of adolescents' emotional problems, coping flexibility, and age; (2) the relationships of adolescents' emotional problems, coping flexibility, and age with stress appraisals (with appraisals measured as perceived threats to relatedness, competence, and autonomy) and adolescents' anticipated use of ways of engagement and disengagement coping; and (3) the associations of stress appraisals with ways of coping. We hypothesized that adolescents higher in emotional problems would appraise more threat and anticipate coping less constructively with stressors, including less ways of engagement coping and more ways of disengagement coping. We hypothesized the opposite associations for coping flexibility, whereby adolescents higher in coping flexibility would appraise less threat and cope more constructively with stressors. Finally, we hypothesized that appraising more threat would be associated with less constructive coping. We were not certain of how age would relate to stress appraisal and ways of coping.

In addition to testing the above hypotheses, a secondary aim was to explore how the stressor domain might impact on these associations. Thus, we first tested a general model considering appraisals and ways of coping as latent variables indicated by measures of appraisal and coping, respectively, which were specific to each of the three salient domains of stress during adolescence—academic, parent, and peer. We then tested a second model to quantify associations of stress appraisals with ways of coping within each stressor domain.

## 2. Materials and Methods

### 2.1. Participants and Procedure

Participants ranged from 10 to 15 years of age ($M$ = 12.5, $SD$ = 1.5), with 195 boys (48%), 208 girls (50%), and 7 (2%) nonbinary/other (total $N$ = 410). Adolescents could select multiple choices from the following ones to describe themselves: White, Asian, Australian First Peoples, Torres Strait Islander or Pacific Islander, other race/ethnicity, born in Australia, and born in New Zealand. Overall, 99% of adolescents ticked at least one answer, with 65% reporting White, 9% Asian, 5% Australian First Peoples, Torres Strait Islander or Pacific Islander, 37% other race/ethnicity, 85% born in Australia, and 7% born

in New Zealand. Students attended grades 7 to 10 at three Australian secondary schools (ages 13–15, 56%) or were in grades 5 or 6 at five of their feeder schools (ages 10–12, 44%). School websites reported that 14–29% of students fell into the lowest and 4–30% into the highest income quartiles, whereas the proportion of students who spoke a language other than English at home ranged from 5–29%.

The participants in this study were drawn from a longitudinal study that had been initiated (T1) six months previously. Students were selected to be contacted about participation in the present study from among 677 parents (78%) who had agreed to be contacted for other research studies. The aim was to achieve a random 50% participation rate in the present study, with the expectation that this would provide a high representation of the larger sample. The randomly selected students were contacted via phone or email by research assistants to gain additional consent for participation. Parents and students were introduced to the study protocol and provided with a link to an online survey with embedded film clips. Once assenting to participate, adolescents were recontacted up to seven times via phone or email to remind them about survey participation. Data were collected in two periods as follows: July–October 2020 and March–June 2021 to avoid data collection at the end of the school year and in the Australian summer break.

All aspects of this study were approved by the Griffith University Human Research Ethics Committee (protocol # 2019-178) prior to contacting schools and parents. The study was carried out in accordance with the World Medical Association Declaration of Helsinki. The datasets generated during and/or analyzed for the current study are available from the corresponding author on reasonable request.

### 2.2. Measures

### 2.2.1. Video Excerpts

Adolescents viewed six short (<30 s) film clips of stressors involving academics/school (2 clips), parents (2 clips), and peers (2 clips). The stressful scenes depicted were as follows: a girl suspected of cheating by a teacher at school, a girl finding out that she did much worse than she expected on a school written assignment, a boy having a verbal argument with his father, a girl witnessing a loud argument between her parents, a boy who was the last one picked for a team, and a boy being teased and laughed at by classmates. A positive scene of a group at a birthday party was shown mid-point to distract from the patterns of stressors. All the stressors employed are ones that are commonly experienced and have been found to be distressing for adolescents [11–13]. The film clips were approximately 30 s in length and depicted an adolescent appearing close in age to the participants as the central figure. All scenes were in English and were from general (G)-rated films or YouTube. This analogue procedure was developed in multiple previous studies that involved numerous pilot tests of written vignettes and film clips, which were followed by refining the final collection of vignettes and film clips to portray daily hassles (rather than major stressful life events, e.g., loss of a parent) and events that were reported to be commonly experienced by adolescents [28,29,49,50]. Students completed a series of questions after they viewed each film clip. For the six stressors, they reported the following points.

### Stress Appraisals

Stress appraisals were measured as perceived threat to relatedness, competence, and autonomy. Participants were first reminded of the stressor (e.g., IF YOU WERE SUSPECTED OF CHEATING LIKE THIS) followed by questions to measure the three threats. To measure the threat to relatedness, participants were asked the question as follows: how much would you feel. . .rejected or excluded? To measure the threat to competence, participants were asked the question as follows: how much would you feel. . . dumb or stupid? To measure the threat to autonomy, participants were asked the question as follows: how much would you feel. . . bossed around or coerced? Responses ranged from 1 (No, not at all) to 5 (Yes, very much). The appraised need threats were highly intercorrelated within each stressor

domain (all > 0.50), so they were averaged (6 items each) to produce a total appraised threat score for academic (Cronbach's $\alpha$ = 0.70), parent ($\alpha$ = 0.80), and peer ($\alpha$ = 0.84) stressors.

Intended Ways of Coping

Active, withdrawal, self-reliance, and helplessness ways of coping were measured with eight items per video stressor (see Appendix A Table A1 for the items). The responses to each item could range from 1 (No, not at all) to 5 (Yes, very much).

Although designed to capture reports of four engagement and four disengagement ways of coping, the factor structure of all items for each stressor domain were explored using three exploratory factor analyses (EFA; one each for ways of coping with academic, parent, and peer stressors). For each EFA, principal axis factoring with oblique rotation was used. Three of the engagement items (support seeking, problem solving, and information seeking) and three of the disengagement items (social withdrawal, escape, and submission) had high factor loadings on two separate factors in each of the EFAs. However, self-reliance loaded on a third factor and helplessness had moderate loadings on each of the two factors (negative on engagement coping and positive on disengagement coping). Therefore, support seeking, problem solving, and information seeking were averaged to produce total active coping scores for academic (Cronbach's $\alpha$ = 0.80), parent ($\alpha$ = 0.76), and peer ($\alpha$ = 0.77) stressors. Similarly, social withdrawal, escape, and submission were averaged (6 items each) to produce total withdrawal coping scores for academic (Cronbach's $\alpha$ = 0.73), parent ($\alpha$ = 0.77), and peer ($\alpha$ = 0.85) stressors. Self-reliance and helplessness were maintained as separate scores. The correlations between the two self-reliance items were $r$ = 0.38 for academic, $r$ = 0.35 for parent, and $r$ = 0.50 for peer stressors. The correlations between the two helplessness items were $r$ = 0.30 for academic, $r$ = 0.30 for parent, and $r$ = 0.41 for peer stressors.

### 2.2.2. Emotional Problems: Depression and Social Anxiety

As part of the T1 survey completed about 6 months prior to this study, emotional problems were measured with three items from the Children's Depression Inventory (CDI; e.g., I feel sad) [51] and six items from the Social Anxiety Scale for Adolescents (SAS-A; e.g., I worry about doing something new in front of others) [52]. These items were selected for the present study after completing factor analyses of the data collected previously (in 2015) from similar-aged Australian adolescents who had completed the full CDI and SAS-A. The responses to each item could range from 1 (No! Not at all true) to 6 (Yes! Totally true). Depression and social anxiety items were first averaged to produce two subscale scores, which were then averaged to produce a total score, with a higher score indicating more problems (Cronbach's $\alpha$ = 0.93).

### 2.2.3. Coping Flexibility

As part of the T1 survey completed about 6 months prior to this study, participants were prompted with a definition of stress and coping before completing 6 items from the Self-Perceived Flexible Coping Scale (SFCS; e.g., "I can come up with lots of ways to make myself feel better if I am stressed") [48]. Item responses ranged from 1 (No! Not at all true) to 7 (Yes! Totally true). Total scores were created by averaging the relevant items so that higher scores represented a higher level of coping flexibility (Cronbach's $\alpha$ = 0.91).

### 2.3. Overview of the Data Analyses

There were no missing data. Pearson's correlations were estimated including all the measures before conducting the main analyses involving fitting latent variable structural equation models (LVSEMs) to test all the associations across domains. In all the LVSEMs, latent variables were estimated for stress appraisals and the four ways of coping, with each latent variable having three indicators as follows: one each for measures for the academic, parent, and peer stressors. We freed all the significant correlations between emotional problems, coping flexibility, and age; we freed all the direct paths from emotional problems, coping flexibility, and age to latent stress appraisal and latent ways of coping; and we

freed all the direct paths from latent stress appraisal to latent ways of coping. Finally, if significant, covariation between the latent ways of coping were freed. In addition to reporting the chi-square to judge the model fit, we also report the CFI and RMSEA, where a good fit is indicated thorough a CFI of >0.90 and RMSEA of <0.08.

Next, in a second model, we freed the effects of stress appraisal on coping within stressor domains (i.e., stressor-specific associations) to test whether these paths were significant and improved the model fit relative to the first model (tested with the $\chi^2$-difference test). In this way, we draw conclusions about the cross-domain associations of stress appraisals and ways of coping, as well as those considering how the stress appraisal that is specific to each stressor domain (academic-, parent-, or peer-related) relates to adolescents' intended ways of coping. Finally, we fit three models that were compared with the fit of the second model to test whether (1) emotional problems, (2) coping flexibility, and (3) age are effectively modeled as trait-level variables because they have a similar influence on stress appraisal and ways of coping across academic, parent, and peer stressors or whether associations differ by stressor domain.

## 3. Results

### 3.1. Descriptive Statistics and Correlations between Measures

Table 1 provides the means (*M*s) and standard deviations (*SD*s) for measures of emotional problems, coping flexibility, and age as well as correlations of these three measures with all stress appraisals and ways of coping. The emotional problem score was significantly positively correlated with stress appraisal in each stressor domain, and with less active coping with parent and peer (but not academic) stress. The emotional problem score was also strongly positively correlated with withdrawal coping for each stressor domain and positively associated with self-reliance and helplessness for each stressor domain. The associations of coping flexibility with stress appraisal and ways of coping tended to be the converse of the associations for emotional problems (and emotional problems and coping flexibility were significantly negatively correlated). Finally, older adolescents anticipated using less active coping with parent and peer stress and more self- reliance (especially for academic and peer stressors) and helplessness to cope with each stressor domain. Older adolescents also scored higher in emotional problems. Table 2 provides the *M*s and *SD*s for all the measures of stress appraisal and ways of coping, as well as the Pearson correlations between these measures. Stress appraisal was positively correlated with anticipating more use of all the ways of coping. Peer stress was appraised as more threatening than academic or parent stress, with the Paired $t(409) = 3.07$ and 3.59, respectively, with $p < 0.001$ for each.

**Table 1.** Correlations of adolescents' emotional problems, coping flexibility, and age with stress appraisals and ways of coping (*N* = 410).

| | Emotional Problems | Coping Flexibility | Age |
|---|---|---|---|
| Academic stress appraisal | 0.34 *** | −0.12 * | 0.04 |
| Parent stress appraisal | 0.30 *** | −0.12 * | 0.01 |
| Peer stress appraisal | 0.25 *** | 0.05 | −0.08 |
| Academic active coping | −0.07 | 0.24 ** | 0.04 |
| Academic withdrawal coping | 0.44 *** | −0.22 ** | 0.00 |
| Academic self-reliance coping | 0.18 *** | −0.10 * | 0.28 *** |
| Academic helplessness coping | 0.19 *** | −0.20 ** | 0.11 * |
| Parent active coping | −0.17 *** | 0.22 ** | −0.11 * |
| Parent withdrawal coping | 0.44 *** | −0.22 ** | 0.03 |
| Parent self-reliance coping | 0.17 *** | −0.09 | 0.11 * |
| Parent helplessness coping | 0.26 *** | −0.20 ** | 0.12 * |
| Peer active coping | −0.19 *** | 0.27 ** | −0.10 * |
| Peer withdrawal coping | 0.40 *** | −0.15 ** | −0.02 |
| Peer self-reliance coping | 0.13 * | −0.03 | 0.23 *** |
| Peer helplessness coping | 0.36 *** | −0.22 ** | 0.15 ** |
| Emotional problems | -- | −0.37 *** | 0.13 ** |
| Coping flexibility | | -- | 0.01 |
| *M* | 2.94 | 3.23 | 12.50 |
| *SD* | 1.43 | 1.40 | 1.50 |

* $p < 0.05$. ** $p < 0.01$. *** $p < 0.001$.

**Table 2.** Correlations between stress appraisals and ways of coping ($N$ = 410).

| | | 1 | 2 | 3 | 4 | 5 | 6 | 7 | 8 | 9 |
|---|---|---|---|---|---|---|---|---|---|---|
| 1 | Acad stress appraisal | | | | | | | | | |
| 2 | Parent stress appraisal | 0.67 *** | | | | | | | | |
| 3 | Peer stress appraisal | 0.61 *** | 0.59 *** | | | | | | | |
| 4 | Acad active C | 0.31 *** | 0.21 *** | 0.33 *** | | | | | | |
| 5 | Acad withdrawal C | 0.63 *** | 0.53 *** | 0.43 *** | 0.04 | | | | | |
| 6 | Acad self-reliance C | 0.11 * | 0.18 ** | 0.03 | 0.05 | 0.15 ** | | | | |
| 7 | Acad helplessness C | 0.12 * | 0.16 ** | 0.05 | −0.40 *** | 0.33 *** | 0.13 ** | | | |
| 8 | Parent active C | 0.17 ** | 0.20 *** | 0.34 *** | 0.52 *** | 0.00 | 0.00 | −0.25 *** | | |
| 9 | Parent withdrawal C | 0.52 *** | 0.63 *** | 0.50 *** | 0.10 * | 0.68 *** | 0.17 ** | 0.28 *** | 0.00 | |
| 10 | Parent self-reliance C | 0.14 ** | 0.25 *** | 0.15 ** | −0.02 | 0.12 * | 0.45 *** | 0.21 *** | 0.05 | 0.22 *** |
| 11 | Parent helplessness C | 0.10 * | 0.15 ** | 0.08 | −0.19 ** | 0.30 *** | 0.16 ** | 0.48 *** | −0.29 *** | 0.40 *** |
| 12 | Peer active C | 0.18 ** | 0.13 * | 0.36 *** | 0.55 *** | −0.01 | −0.06 | −0.27 *** | 0.67 *** | 0.03 |
| 13 | Peer withdrawal C | 0.53 *** | 0.52 *** | 0.61 *** | 0.10 * | 0.69 *** | 0.08 | 0.24 *** | 0.04 | 0.71 *** |
| 14 | Peer self-reliance C | 0.13 ** | 0.15 ** | 0.14 ** | 0.07 | 0.09 | 0.43 *** | 0.17 ** | 0.06 | 0.13 * |
| 15 | Peer helplessness C | 0.25 *** | 0.28 *** | 0.25 *** | −0.13 * | 0.44 *** | 0.16 ** | 0.44 *** | −0.20 *** | 0.45 *** |
| | *M* | 2.38 | 2.37 | 3.29 | 3.16 | 2.20 | 2.80 | 2.08 | 2.64 | 2.64 |
| | *SD* | 0.90 | 0.94 | 1.10 | 0.95 | 0.93 | 1.07 | 0.99 | 1.00 | 1.03 |

| | | 10 | 11 | 12 | 13 | 14 | 15 |
|---|---|---|---|---|---|---|---|
| 1 | Acad stress appraisal | | | | | | |
| 2 | Parent stress appraisal | | | | | | |
| 3 | Peer stress appraisal | | | | | | |
| 4 | Acad active C | | | | | | |
| 5 | Acad withdrawal C | | | | | | |
| 6 | Acad self-reliance C | | | | | | |
| 7 | Acad helplessness C | | | | | | |
| 8 | Parent active C | | | | | | |
| 9 | Parent withdrawal C | | | | | | |
| 10 | Parent self-reliance C | | | | | | |
| 11 | Parent helplessness C | 0.16 ** | | | | | |
| 12 | Peer active C | −0.06 | −0.23 *** | | | | |
| 13 | Peer withdrawal C | 0.14 ** | 0.29 *** | 0.07 | | | |
| 14 | Peer self-reliance C | 0.51 *** | 0.17 ** | 0.05 | 0.01 | | |
| 15 | Peer helplessness C | 0.21 *** | 0.56 *** | −0.28 *** | 0.55 *** | 0.08 | |
| | *M* | 2.66 | 2.26 | 2.70 | 2.64 | 2.82 | 2.39 |
| | *SD* | 1.16 | 1.05 | 1.03 | 1.18 | 1.22 | 1.14 |

Note. Acad = academic. C = coping. * $p < 0.05$. ** $p < 0.01$. *** $p < 0.001$.

### 3.2. Across Stressor LVSEM Model

The first model testing the associations of emotional problems, coping flexibility, and age with stress appraisal and ways of coping across stressor domains had a less than adequate fit to the data, with $\chi^2(113) = 484.37$, $p < 0.001$, CFI = 0.88, RMSEA = 0.090 [90% CI = 0.082 to 0.098], and $p < 0.001$. However, many associations were moderate to strong and significant; the results of the measurement and structural parts of the model are shown in Figure 1. As can be seen, when emotional problems and coping flexibility were simultaneously considered, the emotional problem score was positively associated with appraising more need threat and had a direct negative association with active coping intentions and direct position associations with withdrawal and helplessness coping. In comparison and not as hypothesized, coping flexibility was not significantly associated with stress appraisal, but it had unique, direct associations with more active coping and less helplessness. Age was associated with anticipating more self-reliance to cope with stressors but also with anticipating more helplessness. Finally and unexpectedly, appraising more threat from stressors was associated with anticipating more use of all the ways of coping, with particularly strong associations between stress appraisal and withdrawal coping. Age was directly associated with anticipating more self-reliance to cope with stressors but also more anticipated helplessness.

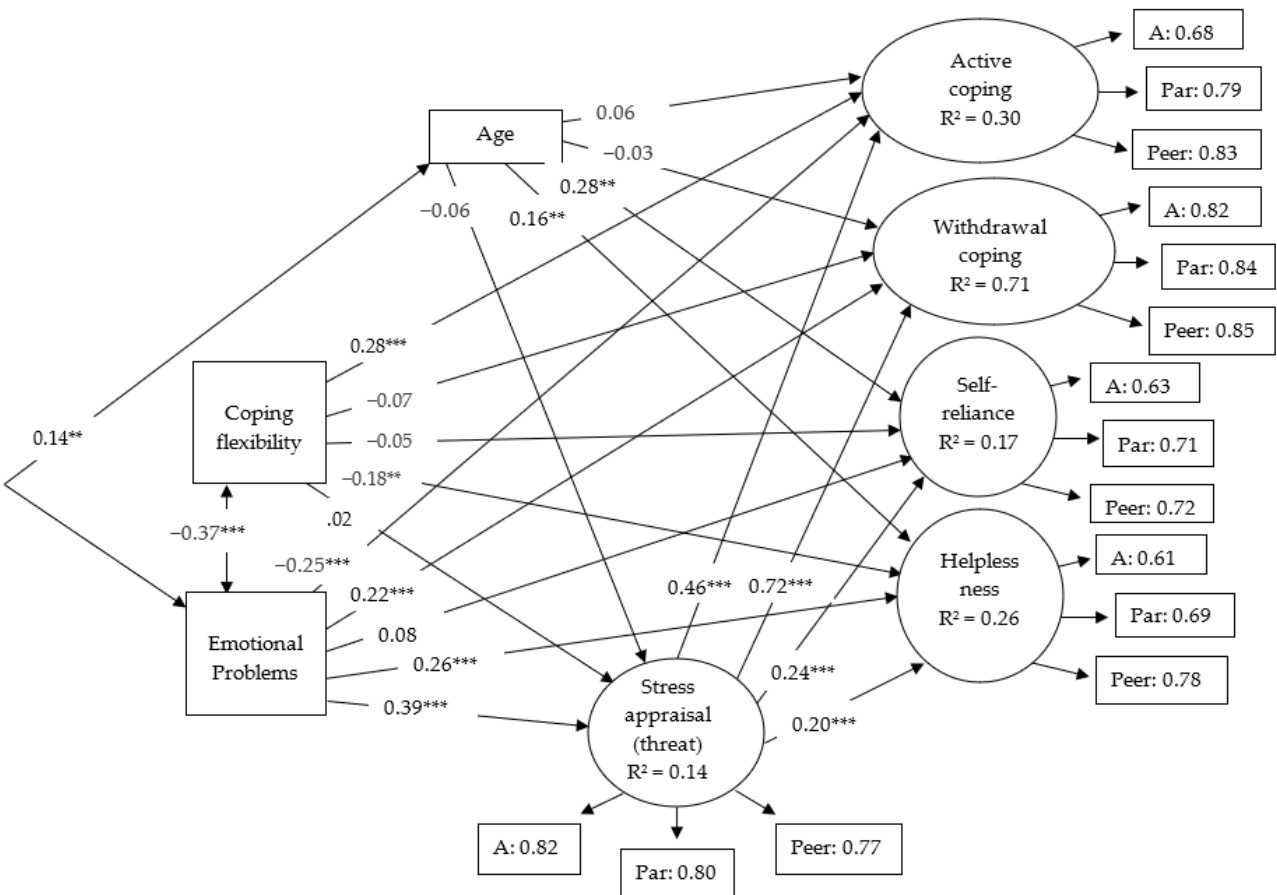

**Figure 1.** Results of the LVSEM testing associations of emotional problems, coping flexibility, age, psychological need threat appraisals, and ways of coping across three stressor domains (A: academic; Par: parent; Peer). $\chi^2(113) = 484.37$, $p < 0.001$, CFI = 0.88, RMSEA = 0.090 [90% CI = 0.082 to 0.098], and $p < 0.001$. Fit improved when domain-specific threat appraisal was linked to domain-specific ways of coping—$\chi^2(101) = 305.08$, $p < 0.001$, CFI = 0.94, RMSEA = 0.070 [90% CI = 0.061 to 0.079], $\Delta\chi^2(12) = 179.29$, and $p < 0.05$. ** $p < 0.01$. *** $p < 0.001$.

### 3.3. Relations of Stress Appraisal with Coping across and within Stressor Domains

In the second model, we freed paths from stress appraisal for each stressor domain (academic, parent, and peer) to the intended ways of coping with that same stressor domain. This "across and within stressor domain model" had a good fit to the data, $\chi^2(101) = 305.08$, $p < 0.001$, CFI = 0.94, RMSEA = 0.070 [90% CI = 0.061 to 0.079], and $p < 0.001$, and the fit was a significant improvement compared with that of the first model, $\Delta\chi^2(12) = 179.29$ and $p < 0.05$. In this model, the stress appraisal specific to each stressor domain had direct associations with anticipating more use of both active and withdrawal coping with the same stressor domain, $\beta$ ranged from 0.16 to 0.26 for active coping and $\beta$ ranged from 0.32 to 0.33 for withdrawal coping, all with $p < 0.001$ (see Table 3). In addition, stress appraisal in response to parent stressors was associated with anticipating more use of self-reliance, with $\beta = 0.16$ and $p = 0.002$, and stress appraisal in response to peer stressors was associated with more helplessness, with $\beta = 0.16$ and $p < 0.001$. The paths from stress appraisal to ways of coping at the latent factor level were weakened in most cases, but the pattern of significant associations was maintained. Emotional problems remained associated with all the ways of coping except that of self-reliance, with $\beta = -0.25$, 0.31, and 0.27 for active, withdrawal, and helplessness coping, respectively, all with $p < 0.001$. Coping flexibility remained significantly associated with more active coping, with $\beta = 0.30$ and $p < 0.001$, and less helplessness, with $\beta = -0.19$ and $p = 0.001$. Age remained positively associated with self-reliance, with $\beta = 0.29$ and $p < 0.001$, and helplessness, with $\beta = 0.16$ and $p = 0.002$.

**Table 3.** Standardized directional path estimates ($\beta$) with associated *p*-values in the model freeing relations of stress appraisal with coping across and within stressor domains.

| Predictor | Outcome | β | *p* |
|---|---|---|---|
| *Effects on latent stress appraisal (appraisal across stressors)* | | | |
| Age | Stress appraisal | −0.05 | 0.313 |
| Emotional problems | Stress appraisal | 0.39 | <0.001 |
| Coping flexibility | Stress appraisal | 0.02 | 0.736 |
| Age | Stress appraisal | −0.05 | 0.313 |
| *Effects on latent ways of coping (coping across stressors)* | | | |
| Emotional problems | Active coping | −0.26 | <0.001 |
| Emotional problems | Withdrawal coping | 0.31 | <0.001 |
| Emotional problems | Self-reliance coping | 0.09 | 0.188 |
| Emotional problems | Helplessness coping | 0.27 | <0.001 |
| Coping flexibility | Active coping | 0.30 | <0.001 |
| Coping flexibility | Withdrawal coping | −0.09 | 0.078 |
| Coping flexibility | Self-reliance coping | −0.04 | 0.463 |
| Coping flexibility | Helplessness coping | −0.19 | 0.001 |
| Age | Active coping | −0.07 | 0.199 |
| Age | Withdrawal coping | −0.04 | 0.435 |
| Age | Self-reliance coping | 0.29 | <0.001 |
| Age | Helplessness coping | 0.16 | 0.002 |
| Stress appraisal | Active coping | 0.27 | <0.001 |
| Stress appraisal | Withdrawal coping | 0.44 | <0.001 |
| Stress appraisal | Self-reliance coping | 0.10 | 0.246 |
| Stress appraisal | Helplessness coping | 0.11 | 0.196 |
| *Effects of stress appraisal for each domain on ways of coping within domain* | | | |
| Academic stress appraisal | Academic active coping | 0.26 | <0.001 |
| Parent stress appraisal | Parent active coping | 0.16 | <0.001 |
| Peer stress appraisal | Peer active coping | 0.18 | <0.001 |
| Academic stress appraisal | Academic withdrawal coping | 0.33 | <0.001 |
| Parent stress appraisal | Parent withdrawal coping | 0.32 | <0.001 |
| Peer stress appraisal | Peer withdrawal coping | 0.32 | <0.001 |
| Academic stress appraisal | Academic self-reliance coping | 0.04 | 0.434 |
| Parent stress appraisal | Parent self-reliance coping | 0.16 | 0.002 |
| Peer stress appraisal | Peer self-reliance coping | 0.08 | 0.112 |
| Academic stress appraisal | Academic helplessness coping | 0.00 | 0.930 |
| Parent stress appraisal | Parent helplessness coping | −0.03 | 0.609 |
| Peer stress appraisal | Peer helplessness coping | 0.16 | <0.001 |

### 3.4. Coping Flexibility: Effects across and within Stressor Domains

In the third model, we built on the second model by freeing paths from coping flexibility to the intended ways of coping for each stressor domain (academic, parent, and peers). This model had a good fit to the data, with $\chi^2(91) = 290.34$, $p < 0.001$, CFI = 0.94, RMSEA = 0.073 [90% CI = 0.064 to 0.083], and $p < 0.001$, but the fit was not a significant improvement compared with that of the previous model, with $\Delta\chi^2(10) = 14.74$ and $p > 0.05$, thereby suggesting that coping flexibility can be considered a trait-level influence associated with stress appraisal and coping across stressor domains.

### 3.5. Emotional Problems: Effects across and within Stressor Domains

In the fourth model, we built on the second model by freeing paths from emotional problems to the ways of coping for each stressor domain (academic, parent, and peers). This model had a good fit to the data, with $\chi^2(91) = 290.31$, $p < 0.001$, CFI = 0.94, RMSEA = 0.074 [90% CI = 0.065 to 0.083], and $p < 0.001$, but the fit was not a significant improvement compared with that of the second model, with $\Delta\chi^2(10) = 14.77$ and $p > 0.05$, thereby suggesting that emotional problems can be considered a trait-level influence associated with stress appraisal and coping across stressor domains.

### 3.6. Age: Effects across and within Stressor Domains

In the final model, we freed paths from age to the intended ways of coping for each stressor domain (academic, parent, and peers). This model had a good fit to the data, with $\chi^2(91) = 271.50$, $p < 0.001$, CFI = 0.94, RMSEA = 0.070 [90% CI = 0.060 to 0.070], and $p < 0.001$, and the fit was a significant improvement compared with the second model, with $\Delta\chi^2(10) = 33.58$ and $p < 0.05$, thereby suggesting that age effects can differ across stressor domains. Age had direct associations with less active coping with peer stress only, with $\beta = -0.10$ and $p = 0.035$. Age was associated with more self-reliance with academic stress, with $\beta = 0.27$ and $p < 0.001$, and peer stress, with $\beta = 0.22$ and $p < 0.001$. Moreover, age was associated with more helplessness with parent stress, with $\beta = 0.10$ and $p = 0.037$, and peer stress, with $\beta = 0.14$ and $p = 0.003$. Age was not associated with withdrawal coping in any domain.

## 4. Discussion

Although there are environmental events that can be identified as objectively stressful for adolescents, such as failing an exam, being rejected by peers or arguing with a parent, we expected in this study that adolescents would differ in their stress appraisals, which would explain not only their ways of coping with stress but also that the associations of stress appraisals and the ways of coping may differ between the three stressor domains of academics, peers, and parents. One novel feature of this study was the operationalization of stress appraisals as adolescents' perceptions of the threat to their relatedness, competence, and autonomy, as well as the measurement of anticipated ways of engagement and disengagement coping within the three separate domains of academic, parent, and peer stressors. We focused on these three domains because adolescents have reported these as some of the most common and salient stressors in their lives [11–13], and appraisals and ways of coping in each domain (and their foundations in emotional problems, coping flexibility, and age) could differ. The present study investigated these associations, testing associations in general (not considering the domains specifically) and within each stressor domain to acknowledge that ways of coping can be constrained or supported to some extent by the particular domain of stress under consideration [2,3]. There were four general findings of this study, which we consider in the sections below.

### 4.1. Stress Appraisals and Ways of Coping at the Trait Level

The first general finding was that appraisals of threat from stressful events were reliably measured by building on previous analogue techniques [31,49,50]. Although there can be stressful events that primarily challenge physical safety and security, many—probably

all—stressful events interfere with the psychological needs of relatedness (i.e., belonginess, connection, and closeness such as loss, interpersonal conflict, or rejection), competence (i.e., feeling effective and competent such as during an experience of failure), and/or autonomy (i.e., feeling able to have choice and agency such as by following strict rules). These threats determine when environmental events are more distressing (i.e., are stressors) and initiate coping. Despite this possibility, most research on stress appraisals has focused on controllability, blame, or perceived access to resources, which are important appraisals that link to coping (e.g., in [28,46,53–55]).

Stress appraisal also was linked to adolescents' ways of coping with stressors, with a higher level of appraised threat associated with intended reliance on more of all the ways of coping. Thus, adolescents who appraised more threat from the presented stressors intended to use more active ways of coping (e.g., problem solving and support seeking), which are often described as adaptive, and they reported higher intentions to be self-reliant and to respond by withdrawing or feeling helpless. Thus, stress appraisal was related to more, not less, intention to use adaptive ways of coping (i.e., active coping and self-reliance), in contrast to what was hypothesized based on developmental theories of coping. Nevertheless, these findings are consistent with past research that shows positive associations between the level of distress (e.g., sadness, worry, and anger) from stressful events and all the ways of coping [2] but differs from the findings of the appraisal of controllability using analogue methods. Zimmer-Gembeck and Skinner [2] reported positive a association of controllability with a composite of coping that included active coping, self-reliance, and support seeking but a negative association of controllability with a composite of coping that included isolation/withdrawal and helplessness. Future research should be conducted to consider stress appraisal by measuring threats to psychological needs alongside other more commonly studied appraisals, such as controllability, and emotional reactions to consider how multiple types of appraisals are foundational for ways of coping and mental health among adolescents. Such information would be particularly useful for interventions with young people presenting high levels of depressive and social anxiety symptoms who may need support to understand their psychological needs and how perceived threats to relatedness, competence, and autonomy yield stress responses as well as support in addressing how the controllability of stressors in their lives might be important to consider before identifying constructive ways of coping.

### 4.2. Stress Appraisals and Ways of Coping within Stressor Domains

Associations between stress appraisal and ways of coping were found when modeled across domains of stress, thereby identifying associations at more of a "trait" level. Yet, a second general finding was that considering associations within the stressor domain (academic, parent, and peer domains) resulted in a better model fit. This supports the value in modeling person-level associations between stress appraisal and ways of coping but also suggests that there is added predictive value in considering specificity within stressor domains in future research. In particular, the clearest differences were found for the associations of parent stress appraisals with higher self-reliance and peer stress appraisals with greater helplessness, whereby these specific associations did not extend to academic or to the other social stressor domain. These findings suggest that adolescents have general ways of appraising and then coping with stress, but that they can also identify the opportunities and constraints they are afforded within different stressor domains, as has been described in developmental coping theory [8]. For example, adolescents should be able to understand the different threats and the different ways of coping most useful when responding to social vs. nonsocial stressors or when responding to authority figures (such as parents or teachers) vs. peers. Overall, there are domain–general—or what might be referred to as trait-level—ways through which adolescents appraise threat from stressors and this relates to reports of intended ways of coping, but these appraisals and ways of coping could differ depending on the stressor domain. Such a finding is important for methodologies in the future by suggesting that the stressor domain is relevant. Findings from a study

of the influence of appraisals and coping, or of how appraisals and coping may develop with age, conducted within one stressor domain (e.g., academic workload stress) may not be directly generalized to other stressor domains (e.g., peer victimization). Similarly, a study that does not identify the stressor domain may not be directly generalizable to a specific stressor domain. Even the focus on academic, parent, and peer stressors in the present study may be too coarse, masking differences between appraisals and ways of coping that could vary between different reasons for stress (e.g., academic workload vs. teacher–student relationships or friendship conflict vs. peer rejection).

When stressor domains were compared, it is notable that adolescents appraised the highest level of threat after viewing peer stressors. Threat may be highest for peer relationship stressors because peers rapidly ascend in importance for belonging, support, and companionship during this period of life. This importance may have been reflected in the high appraised threat from stressors that involved lack of inclusion and ridicule by peers [56–58]. On the other hand, it is possible that other features of the film clips could explain this difference. For example, peer relationship stressors could be perceived as less controllable. In addition, both film clips of peer stressors not only portrayed the lack of inclusion or ridicule by one peer but also portrayed witnesses (as would occur in real life) [50]. Although one academic stressor included witnesses to the stressor (being caught cheating), all the other film clips portrayed only the individual adolescent or only the adolescent and a parent or parents.

### 4.3. Emotional Problems, Coping Flexibility, and Age

A third general finding was that the trait level differences in stress appraisal were predicted by adolescents' emotional problems (depressive and social anxiety symptoms) but not by coping flexibility or age. However, emotional problems and coping flexibility each had unique associations with ways of coping. Adolescents reporting more emotional problems reported that they would cope with stressors less constructively—they intended to use less active coping and more withdrawal and helplessness. Adolescents higher in coping flexibility reported they would cope with stressors in more constructive ways—they intended to use more active coping and to be less helpless. Although these findings were somewhat different than hypothesized (we expected that adolescents' emotional problems and coping flexibility would each uniquely relate to stress appraisal and ways of coping), they do confirm and extend past research [4] that has identified the wide-reaching impact of emotional problems on the entire stress–appraisal–coping process through heightened perceptions of threat.

It was surprising that coping flexibility and adolescents' age did not have unique associations with stress appraisal in the academic, parent, or peer stressor domains. One possibility is that most of the associations of coping flexibility and age with stress appraisal are captured in the association between emotional problems and stress appraisal because emotional problems did have negative and positive associations with coping flexibility and age, respectively. However, the zero-order correlations suggest that this is not an explanation given that even they showed low and mostly nonsignificant associations of coping flexibility and age with stress appraisal (for each of the academic, parent, and peer domains). Another explanation arises from considering how coping flexibility was measured as beliefs in the capacity to draw on multiple ways to cope with stress. Thus, coping flexibility reflected beliefs about the self, and it is possible that these self-beliefs do not play a direct role in stress appraisal given that appraisal of environmental stress may be derived from features of the stressors themselves rather than being dependent on self-beliefs. These findings provide ideas on where to focus prevention and intervention programs to improve adolescents' coping with stress; it seems important to concentrate efforts both on self-beliefs about being able to cope with stress and on techniques to reduce need-related sensitivities to stressful events.

The associations of emotional problems and coping flexibility were found when modeled across stressor domains, but there was no improvement to model the fit after

adding paths to tests links within each separate stressor domain. This suggests that emotional problems and coping flexibility have cross-domain effects on stress responding, making it even more important to consider them as areas to spotlight in prevention and intervention programs.

### 4.4. Adolescent Age, Self-Reliance, and Helplessness

The fourth general finding regards adolescents' age as a unique correlation of stress appraisal and ways of coping. Adolescents' age, ranging from 10 to 15 years, was the only one of the three measured trait level variables that had an association with self-reliance, with older adolescents reporting intentions to be more self-reliant but also reporting that they felt they would respond to stressors with more helplessness. Furthermore, age was the only one of these three trait level variables that showed some additional predictive value at the level of the stressor domain, revealing that the age-related increase in self-reliance may be most clear when appraising more threat from parent stress and the age-related increase in helplessness may be most clear when appraising more threat from peer stress. It would be very valuable in future research to clarify the stressor domain when studying age-related change in ways of coping (i.e., the development of coping).

### 4.5. Study Limitations

This study is not without certain limitations. First, stressors were portrayed using standardized film clips and adolescents reported their intended (not actual or recalled) ways of coping. We used these film clips in order to have some control over the stressful stimuli and to gather stress appraisals and intended ways of coping when all adolescents viewed three different types of stressors (academic, parent, and peer). There is evidence that intended ways of coping do converge with remembered ways of coping with experienced stressors [59], but it is not yet known if stress appraisals measured as threats to psychological needs would converge with similar stress appraisals collected following experienced stressful events. Second, ways of coping were measured with only one item per film clip (two items in total per subscale per domain). However, these measures had high interitem correlations, and this approach has been used successfully in past research [31,49,50]. Third, all the participants viewed the same film clips, and the clips were not randomized. Therefore, boys viewed some clips with a girl as the central character and vice versa. This was done to maintain standardization, but this could have affected the data collected either if this interfered with immersion in the content or there were order effects that were undetected. Finally, emotional problems were measured using items drawn from more comprehensive but widely used measures of depression and social anxiety. Although these items were selected based on analyses that supported their representativeness of the measures in their entirety, combining depressive and social anxiety symptoms into a single measure meant we could not draw conclusions about the effects of depressive symptoms compared with the effects of social anxiety symptoms on stress appraisal and ways of coping. Moreover, both emotional problems and coping flexibility were measured about six months prior to the measurement of stress appraisal and coping. There is a possibility that this introduced some errors given that emotional problems and flexibility could have changed over the six months. It is not clear how this could have influenced the findings reported here, but it is most likely that this would have reduced the size of the associations rather than inflating them.

## 5. Conclusions

The purpose of this study was to identify how adolescents' emotional problems, coping flexibility, and age can shape stress appraisals and ways of coping across three major stressor domains for adolescents—academics, interactions with parents, and peer relationships. Incorporating all the measures into latent structural equation models, we also tested the associations of stress appraisals with ways of coping. Using videos of stressful events as stimuli, adolescents appraised six stressors, two in each domain, using items that

reflected threats to relatedness, competence, and autonomy. Adolescents also reported their anticipated ways of coping with each stressor, which were collapsed into four composite ways of coping as follows: two positive ways of engagement coping labeled as active coping (i.e., support seeking, problem solving, and information seeking) and self-reliance, and two negative ways of disengagement coping labeled as withdrawal coping (i.e., social withdrawal, escape, and submission) and helplessness. This study was the first to measure stress appraisal as a perceived threat to the psychological needs of relatedness, competence, and autonomy, separately measuring appraisals of portrayals of academic, parent, and peer stressors. It was also the first to model how adolescents' emotional problems (combined depressive and anxiety symptoms), coping flexibility, and age relate to stress appraisal and the intended ways of engagement and disengagement coping both across and within the three stressor domains.

Stress appraisal, when measured as the perception of threat to relatedness, competence, and autonomy, did relate to the ways that adolescents cope with stress. At the general level considering all the stressor domains, adolescents who appraised more threat reported they would use more withdrawal to cope and would feel more helpless, but they also reported they would use more active coping and that they want to be more self-reliant. When the domain of stress was considered, some of the associations were stronger within certain domains relative to others, and two associations stood out as different, with there being a positive effect of stress appraisal on self-reliance only when the stressor was in the parent–adolescent relationship domain, and the positive effect of stress appraisal on helplessness was only significant when the stressor was in the peer relationship domain. Adding to these associations, adolescents' emotional problems, coping flexibility (belief in the capacity to use multiple ways to cope with stress when needed), and age had influence, with emotional problems having the most far-reaching effects on stress appraisal (more appraisal of threat) and less ways of engagement coping and more ways of disengagement coping (less active and more problem coping). However, coping flexibility related to more use of ways engagement of coping, and age was uniquely related to more self-reliance and helplessness.

Taken together, the present study's findings suggest that interventions for adolescents, which are closely timed to the experience of a stressful event, could be useful for growth by supporting youth to consider how appraisals can interfere with relatedness, competence, and autonomy (e.g., feeling rejected or excluded, incompetent, or coerced), to consider alternative appraisals, and to understand how appraisals relate to the many available ways that are available to cope with stress in the short and in the longer term. This support would acknowledge how coping actions can be relied upon to solve problems and regulate emotion while also helping adolescents to reflect on their own beliefs and behaviors when facing stressful events (e.g., for more details see [3,60,61]). At the same time, the findings suggest that adolescents could benefit from discussions and education regarding coping with stressors to support their development of coping flexibility. This could be supported by providing ongoing opportunities to practice and receive direct feedback on stress appraisals and ways of coping from parents and teachers or other professionals when hassles, challenges, or other stressors occur at home or at school.

**Author Contributions:** Conceptualization, M.J.Z.-G. and E.A.S.; methodology, M.J.Z.-G.; formal analysis, M.J.Z.-G.; investigation, M.J.Z.-G. and E.A.S.; data curation, M.J.Z.-G.; writing—original draft preparation, M.J.Z.-G.; writing—review and editing, M.J.Z.-G. and E.A.S.; project administration, M.J.Z.-G.; funding acquisition, M.J.Z.-G. and E.A.S. All authors have read and agreed to the published version of the manuscript.

**Funding:** This work was supported by the Australian Research Council (DP190101170).

**Institutional Review Board Statement:** The study was approved by the Griffith University Human Research Ethics Committee (GU Ref No: 2019/178). The study was carried out in accordance with the World Medical Association Declaration of Helsinki.

**Informed Consent Statement:** Written informed consent was obtained from the parents of all the participants.

**Data Availability Statement:** The data are available upon reasonable request from the first author.

**Acknowledgments:** We thank Kathryn Modecki, Amanda Duffy, Allison Waters, Lara Farrell, and David Shum for their input into the larger project from which these data were drawn. We also thank Alex Gardner and Tanya Hawes for their project and data management expertise, and we thank Kathy Ryan for statistics on mental health problems around the world. We also thank the students and the schools for their willingness to participate during times of uncertainty and change, and we gratefully acknowledge the important contributions from $N = 24$ research assistants who were critical to the data collection.

**Conflicts of Interest:** The authors report no competing or conflicts of interest.

## Appendix A

**Table A1.** Ways of coping items: academic stressor example.

| IF YOU WERE SUSPECTED OF CHEATING LIKE THIS (CIRCLE ONE number for each item) | NO, Not at All | A Little or Some | In the Middle | Quite a Bit | YES, Very Much |
|---|---|---|---|---|---|
| What would you do? WOULD YOU… | | | | | |
| 1. deal with the situation on your own? (self-reliance) | 1 | 2 | 3 | 4 | 5 |
| 2. go to someone (e.g., parent, teacher) for help? (support-seeking) | 1 | 2 | 3 | 4 | 5 |
| 3. do something to fix it? (problem-solving) | 1 | 2 | 3 | 4 | 5 |
| 4. try to get more information or find out what's going on? (information-seeking) | 1 | 2 | 3 | 4 | 5 |
| 5. go off to be by yourself or really want to be alone? (social withdrawal) | 1 | 2 | 3 | 4 | 5 |
| 6. do nothing? (helplessness) | 1 | 2 | 3 | 4 | 5 |
| 7. get away as fast as you could? (escape) | 1 | 2 | 3 | 4 | 5 |
| 8. give in or give up? (submission) | 1 | 2 | 3 | 4 | 5 |

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
