# Peer review of "Stress Appraisals and Coping across and within Academic, Parent, and Peer Stressors: The Roles of Adolescents’ Emotional Problems, Coping Flexibility, and Age"

_adolescents, doi:10.3390/adolescents4010009_

Round 1

Reviewer 1 Report

Comments and Suggestions for Authors

Thank you for the opportunity to review the manuscript titled “Psychological Need Threat and Coping Across and Within Academic, Parent, and Peer Stressors: The Roles of Adolescents' Emotional Problems, Coping Flexibility, and Age”. he manuscript presents the results of a cross-sectional study on the relationship between emotional problems, coping flexibility, age, psychological need threat, and coping. Overall, I see significant value in the manuscript, and I believe that the results of this study can make an important contribution to the understanding of why adolescents rely on certain coping skills in specific areas of their lives, serving as a crucial step in prevention and intervention research. I have outlined several suggestions below to improve the manuscript.

1)     Given the cross-sectional design of the paper, it is not possible to make statements about the direction of the relationships. While the authors propose a hypothesized model suggesting the direction of associations, such as from threat appraisal to ways of coping, there is evidence in the literature that the way one copes with a situation plays a role in the degree of need satisfaction or frustration. A theoretical model in which psychological basic needs are modeled as both predictors (perception of threat) and outcomes (degree of need frustration) seems to be the most theoretically and empirically sound. This, of course, cannot be tested with the collected data in this paper, but I am curious about the authors' opinion on this.

2)     The manuscript only refers to 'appraisal of threats' and, on lines 80-83, low need fulfillment. The concept of 'need frustration' is not addressed in the manuscript. However, the items to assess 'appraised need threat' seem to clearly refer to need frustration (feelings of exclusion and loneliness ~ relatedness frustration, feelings of incapability ~ competence frustration, and feelings of pressure and coercion ~ autonomy frustration). I would pay sufficient attention in the introduction to how the appraisal of threat relates to need frustration. Is it the same, or is it something different? Is one a predictor, and the other an outcome? (see previous comment).

3)     It is somewhat odd to refer to 'beliefs about coping capacities' at the beginning of the introduction and then suddenly, in the rest of the manuscript, consistently refer to 'coping flexibility.' I would suggest being consistent in the terminology used and talk about coping flexibility from the introduction onwards

4)     Based on the introduction, it is not clear that a model with indirect paths is being tested. This becomes clear only in the 'current study' section. I would specify this earlier and more explicitly.

5)     Relatedly, the three different domains are mentioned only at the end of the introduction for the first time. Why were these three specific domains chosen? What studies have already examined coping with parents, academic, and peer stressors concretely?

6)     While I certainly acknowledge that the examined model is interesting and a valuable contribution to the literature, I would be even more interested in the question of moderation. Does, for example, appraisal of threat only lead to negative coping when there are high emotional problems, and does it lead to adaptive coping strategies when adolescents score high on coping flexibility and/or are older?

7)     The description of the measurement of intended ways of coping is quite complex. Could the order of the description be changed? I suggest ordering it as follows:

I. Could it be specified which overarching constructs the authors aimed to measure? (Was this engagement and disengagement coping?)

II. Which eight ways of coping were measured to achieve these overarching constructs

III. Results of CFA

IV. Final constructs used in analyses: Active, withdrawal, self-reliance, and helplessness

               In addition, a figure depicting the results of the CFA for coping would be helpful.

8)     I was a bit confused that model 3-5 were compared with the first model and not with the second model after concluding that the second model was a better fit for the data. I would suggest comparing model 3-5 to model 2, instead of model 1.

In addition, if possible, I would also show model 2 in an additional figure.

9)     Line 399: why would appraised threats be highest with peer stressors? Are these situations the most recognizable for adolescents, or are these situations more extreme than the academic of parent stressors? Please discuss.

10)  The finding that coping flexibility is unrelated to perceived threat is in line with the dual-process model of self-determination theory. Do the authors think this could be another reason for the lack of an association?

Minor comments:

-        Abstract: Change the sentence “environmental events are distressing because…”. Not all environmental events are distressing. Perhaps change to “environmental events that threaten psychological needs for relatedness, competence, and autonomy are considered distressing for adolescents” or something similar.

-        What is the distribution of age? Was the age of adolescents evenly distributed, or were there, for example, more students from high school than from elementary school?

-        The authors write: "From the original pool of 863 T1 participants in the longitudinal study, 677 parents (78%) had agreed to be contacted for other research studies. Of these, adolescents were randomly selected for recontact." Why were not all 677 families re-contacted?

-        Why was a positive scene of a group at a birthday party shown midway? Please clarify.

-        Relatedly, was the order of the different film clips randomized? Could it be that scenes shown first after the birthday party were more negative or more positively appraised?

-        The subtitle 4.1 gave me the impression that a look within domains was already being taken. I suggest changing this to: "Threat Appraisals and Ways of Coping at the Trait Level."

-        The sentence: 'These self-beliefs may not play a direct role in the appraisal of threat to relatedness, competence, or autonomy because there is a differentiation of individual differences in self-beliefs from individual differences in social perceptions' is rather confusing. Please rewrite.

In conclusion, despite some aspects that need further work, I think the present manuscript has the potential to become an important contribution to the literature on adolescents’ coping.

 I wish you good luck in the further development of the article.

Regards.

Author Response

Reviewer 1

Thank you for the opportunity to review the manuscript titled “Psychological Need Threat and Coping Across and Within Academic, Parent, and Peer Stressors: The Roles of Adolescents' Emotional Problems, Coping Flexibility, and Age”. he manuscript presents the results of a cross-sectional study on the relationship between emotional problems, coping flexibility, age, psychological need threat, and coping. Overall, I see significant value in the manuscript, and I believe that the results of this study can make an important contribution to the understanding of why adolescents rely on certain coping skills in specific areas of their lives, serving as a crucial step in prevention and intervention research. I have outlined several suggestions below to improve the manuscript.

1)     Given the cross-sectional design of the paper, it is not possible to make statements about the direction of the relationships. While the authors propose a hypothesized model suggesting the direction of associations, such as from threat appraisal to ways of coping, there is evidence in the literature that the way one copes with a situation plays a role in the degree of need satisfaction or frustration. A theoretical model in which psychological basic needs are modeled as both predictors (perception of threat) and outcomes (degree of need frustration) seems to be the most theoretically and empirically sound. This, of course, cannot be tested with the collected data in this paper, but I am curious about the authors' opinion on this.

***We completely agree with the limitations of a cross-sectional design and that experiences of coping with stress can influence appraisals of stress and how social situations are perceived more generally - including experiencing need satisfaction or frustration. Although we did not directly relate this to need satisfaction or frustration and coping in our intro paragraphs, we attempted to introduce these stress-coping processes that unfold over time at many levels (e.g., episodic time, developmental time) at the start of the paper. However, we also did not want to go into a great amount of detail here because we did not want to be misleading regarding the design of this particular study. We refer the reader to other literature in the first pages of the Introduction that describes developmental coping theory outlining patterns of bidirectional associations unfolding over time, embedded within or founded in individual vulnerabilities and resource availability and context (which also can change over time). For example, see p. 2:

Yet, when stressors are tackled successfully, they can also be foundational for positive development, such as developing a better understanding of the most constructive ways of coping with different types of stressors [3]. Therefore, experiences with stressors and coping are implicated in the development of emotional problems but they are also important for developing coping beliefs and skills, such as an understanding of one’s own ability to cope flexibility and competently with stressful events (coping flexibility). In turn, emotional problems and coping flexibility can have feed forward effects, impacting on future stress appraisals and ways of coping [4,5].

The above ideas follow directly from transactional conceptualizations of stress and coping [6,7] and are foundational to developmental coping conceptualizations, in particular [3,8]. These theories highlight the roles of stress appraisals and ways of coping as founded in personal resources and vulnerabilities (such as emotional problems and coping flexibility). Stress appraisals include individual perceptions of the reasons for and the significance and meaning of encounters with threatening or challenging events within context. Meaning is the information that organizes coping actions, with coping involving, “constantly changing cognitive and behavioral efforts to manage specific external and/or internal demands that are appraised as taxing or exceeding the resources of the person" [9] (p. 141). The specific ways of coping can vary from active or engagement coping to avoidant, withdrawal or disengagement coping [10,11].

2)     The manuscript only refers to 'appraisal of threats' and, on lines 80-83, low need fulfillment. The concept of 'need frustration' is not addressed in the manuscript. However, the items to assess 'appraised need threat' seem to clearly refer to need frustration (feelings of exclusion and loneliness ~ relatedness frustration, feelings of incapability ~ competence frustration, and feelings of pressure and coercion ~ autonomy frustration). I would pay sufficient attention in the introduction to how the appraisal of threat relates to need frustration. Is it the same, or is it something different? Is one a predictor, and the other an outcome? (see previous comment).

***We have modified the content and the terms we used throughout the paper to place a more distinct focus on stress appraisals, while highlighting that we measured stress appraisals as perceived threat to relatedness, competence, and autonomy. We tried to avoid confusing the text by using terms like need threat, frustration, and fulfilment as much as possible.

3)     It is somewhat odd to refer to 'beliefs about coping capacities' at the beginning of the introduction and then suddenly, in the rest of the manuscript, consistently refer to 'coping flexibility.' I would suggest being consistent in the terminology used and talk about coping flexibility from the introduction onwards

***Thank you for this advice - it has helped shape up this part of the paper. We now refer to coping flexibility and define it early in the Introduction (in the first paragraph) and then use coping flexibility throughout the remainder of the paper.  E.g., see p. 2:

The high rates of mental health problems among youth [1] are thought to be partly due to experiences of stressful life events and significant daily hassles [2]. Yet, when stressors are tackled successfully, they can also be foundational for positive development, such as developing a better understanding of the most constructive ways of coping with different types of stressors [3]. Therefore, experiences with stressors and coping are implicated in the development of emotional problems but they are also important for developing coping beliefs and skills, such as an understanding of one’s own ability to cope flexibility and competently with stressful events (coping flexibility). In turn, emotional problems and coping flexibility can have feed forward effects, impacting on future stress appraisals and ways of coping [4,5].

4)     Based on the introduction, it is not clear that a model with indirect paths is being tested. This becomes clear only in the 'current study' section. I would specify this earlier and more explicitly.

***Given the cross-sectional design of this study, we never reported the indirect effects in the tested models. Instead, we describe only the direct effects in our models. We have made sure not to at all suggest that we tested indirect or mediational pathways in the models we report.

5)     Relatedly, the three different domains are mentioned only at the end of the introduction for the first time. Why were these three specific domains chosen? What studies have already examined coping with parents, academic, and peer stressors concretely?

***We mention domain for the first time in the first paragraph of the manuscript and now make sure to highlight this more often throughout the Introduction and remainder of the paper. We do not know of any other paper that has compared these associations across three different stressor domains as was done here.

6)     While I certainly acknowledge that the examined model is interesting and a valuable contribution to the literature, I would be even more interested in the question of moderation. Does, for example, appraisal of threat only lead to negative coping when there are high emotional problems, and does it lead to adaptive coping strategies when adolescents score high on coping flexibility and/or are older?

***We very much appreciate these additional questions that could be addressed with these data. We have considered these associations in another dataset (on a slightly different topic: coping flexibility as a moderator of the association between ways of coping and well-being outcomes), but we do agree moderation effects would be interesting to consider with the current data in a future study. We felt adding this information to this paper would be quite a departure from our present study aims.

7)     The description of the measurement of intended ways of coping is quite complex. Could the order of the description be changed? I suggest ordering it as follows:

  1. Could it be specified which overarching constructs the authors aimed to measure? (Was this engagement and disengagement coping?)
  2. Which eight ways of coping were measured to achieve these overarching constructs

III. Results of CFA

  1. Final constructs used in analyses: Active, withdrawal, self-reliance, and helplessness

            In addition, a figure depicting the results of the CFA for coping would be helpful.

***We have reorganized this section and provide more details. We had conducted three exploratory factor analyses (EFA; one each for academic coping, parent coping, and peer coping), but we agree this was not clear enough. Given that three different EFAs were conducted, we did not report all numbers in tables, but did try to make our approach and results clearer in the text. See p. 11:

2.2.1.2. Intended ways of coping: Active, withdrawal, self-reliance, and helplessness: Engagement and disengagement ways of coping were measured with four items each per video stressor (eight items total per each of the six stressors). See Appendix 1 for the items. The responses to each item could range from 1 (No, not at all) to 5 (Yes, very much).

Although designed to measure engagement and disengagement coping, the factor structure of the eight items for each stressor domain were also explored using three exploratory factor analyses (EFA; one each for ways of coping with academic, parent, and peer stressors). For each EFA, principal axis factoring with oblique rotation was used. Three of the engagement items (support seeking, problem-solving, and information-seeking) and three of the disengagement items (social withdrawal, escape, and submission) had high factor loadings on two separate factors in each of the EFAs. However, self-reliance loaded on a third factor and helplessness had moderate loadings on each of the two factors (negative on engagement coping and positive on disengagement coping). Therefore, support seeking, problem-solving, and information-seeking were averaged to produce total active coping scores for academic (Cronbach’s a = .80), parent (a = .76), and peer (a = .77) stressors. Similarly, social withdrawal, escape, and submission were averaged (6 items each) to produce total withdrawal coping scores for academic (Cronbach’s a = .73), parent (a = .77), and peer (a = .85) stressors. Self-reliance and helplessness were maintained as separate scores. The correlations between the two self-reliance items were .38 for academic, .35 for parent, and .50 for peer stressors. The correlations between the two helplessness items were .30 for academic, .30 for parent, and .41 for peer stressors.

8)     I was a bit confused that model 3-5 were compared with the first model and not with the second model after concluding that the second model was a better fit for the data. I would suggest comparing model 3-5 to model 2, instead of model 1.

***Thank you for catching this error. We had compared to model 2, so have corrected this in the text.

In addition, if possible, I would also show model 2 in an additional figure.

***We had attempted to construct a figure to illustrate all paths tested in Model 2, but we had determined it was too unwieldy to display all of the paths this way. So, overall, we didn’t think this was helpful to include. Instead, we have included Table 3 that lists all directional paths in this model.

9)     Line 399: why would appraised threats be highest with peer stressors? Are these situations the most recognizable for adolescents, or are these situations more extreme than the academic of parent stressors? Please discuss.

***We suspect that these situations may be perceived to be less controllable and both peer stressors included not only the rejection and exclusion by one peer but included witnesses. Although one academic stressor included witnesses to the stressor (being caught cheating), all other film clips portrayed only the individual adolescent or only the adolescent and their parent or parents. We added this sentence on p. 17:

Peer stress was appraised as more threatening than academic or parent stress, paired t = 3.07 and 3.59, respectively, both p < .001.

And added this to the Discussion (p. 24):

When stressor domains were compared, it is notable that adolescents appraised the highest level of threat after viewing peer stressors. Threat may be highest for peer relationship stressors because peers rapidly ascend in importance for belonging, support, and companionship during this period of life. This importance may have been reflected in the high appraised threat from stressors that involved lack of inclusion and ridicule by peers [19, 68-70]. On the other hand, it is possible that other features of the film clips could explain this difference. For example, peer relationship stressors could be perceived as less controllable. In addition, both film clips of peer stressors not only portrayed the lack of inclusion or ridicule by one peer but also witnesses (like what would occur in real life) [62]. Although one academic stressor included witnesses to the stressor (being caught cheating), all other film clips portrayed only the individual adolescent or only the adolescent and their parent or parents.

10)  The finding that coping flexibility is unrelated to perceived threat is in line with the dual-process model of self-determination theory. Do the authors think this could be another reason for the lack of an association?

***This is an interesting idea and raises so many possibilities that were not addressed here. We think we will need to do more work to consider how coping flexibility might relate to need satisfaction, need frustration (and even need dissatisfaction; Reeve et al., 2023), while also considering perceived controllability (and maybe, separately, autonomy support) in relation to each stressor (as we noted in the Discussion). Thus, given that we do not mention this theory, we did not add anything to this paper to raise this idea for now, because it was not readily apparent how coping flexibility might fit within this theory.

Minor comments:

-        Abstract: Change the sentence “environmental events are distressing because…”. Not all environmental events are distressing. Perhaps change to “environmental events that threaten psychological needs for relatedness, competence, and autonomy are considered distressing for adolescents” or something similar.

***This sentence has been changed. Thank you for catching this.

-        What is the distribution of age? Was the age of adolescents evenly distributed, or were there, for example, more students from high school than from elementary school?

***44% were age 10-12 (primary school age) and the remainder (56%) were age 13-15 (secondary school).  In Australia we rarely have middle or junior high - instead we have primary (grades 1 to 6) and secondary (grades 7 to 12) schools.  In addition, the primary schools that feed into the secondary schools are often on the same groups or very close by, so many of the students are familiar with each other throughout the years. We have added this information on p. 8.

-        The authors write: "From the original pool of 863 T1 participants in the longitudinal study, 677 parents (78%) had agreed to be contacted for other research studies. Of these, adolescents were randomly selected for recontact." Why were not all 677 families re-contacted?

***This was simply because of time and resources.

-        Why was a positive scene of a group at a birthday party shown midway? Please clarify.

***To reduce the emotional load on the participants of viewing six stressful events in a row with no break.

-        Relatedly, was the order of the different film clips randomized? Could it be that scenes shown first after the birthday party were more negative or more positively appraised?

***No, we did not randomize the clips and acknowledge this could have been a limitation and had some unknown influence on the results. We have added this as a limitation, see p. 22:

Third, all participants viewed the same film clips and the clips were not randomized. Therefore, boys viewed some clips with a girl as the central character and vice versa. This was done to maintain standardization, but this could have affected some of the data collected or the findings if this interfered with immersion in the content or there were order effects that were undetected.

-        The subtitle 4.1 gave me the impression that a look within domains was already being taken. I suggest changing this to: "Threat Appraisals and Ways of Coping at the Trait Level."

***We have changed the subtitle as recommended.

-        The sentence: 'These self-beliefs may not play a direct role in the appraisal of threat to relatedness, competence, or autonomy because there is a differentiation of individual differences in self-beliefs from individual differences in social perceptions' is rather confusing. Please rewrite.

***This sentence was not clear and has been reworded to say:

Another explanation arises from considering how coping flexibility was measured -- as beliefs in the capacity to draw on multiple ways to cope to stress. Thus, coping flexibility reflected beliefs about the self and it is possible that these self-beliefs do not play a direct role in stress appraisal, given that appraisal of environmental stress may be derived from features of the stressors themselves, rather than dependent on self-beliefs.

In conclusion, despite some aspects that need further work, I think the present manuscript has the potential to become an important contribution to the literature on adolescents’ coping.

 I wish you good luck in the further development of the article.

***We appreciate the helpful suggestions and close reading of this manuscript.

Regards.

Reviewer 2 Report

Comments and Suggestions for Authors

This is an important area of research. The target population of this study is adolescents who are going through a formative and challenging stage of human development. The introduction explains the primary aim was to investigate how emotional problems, coping flexibility, and age may be related to appraised threat to their psychological needs for relatedness, competence and autonomy and importantly how these relate to their anticipation of their coping strategies. In addition the study aimed to investigate appraised need threat and coping strategy in relation to three domains of stress.  A substantial body of data has been analysed and the results reported in detail. The discussion touches on the implications and possible applications. However, the presentation could be amended to attract greater readership and in turn impact.

In the abstract, Give the two aims and the unique aspect of this study. State how the results may benefit the target population.  Explain how the findings might be applied in practice, as well as in future research.  

In the conclusion, state the aims and emphasise the unique element of this study. Summarise the results, and state how the results may benefit the target population. Explain how the findings might be applied in practice, as well as in future research.  

Minor suggestion:

Line 418, the word using might be replaced by useful: "....would be particularly useful for interventions with...."

Author Response

Reviewer 2

This is an important area of research. The target population of this study is adolescents who are going through a formative and challenging stage of human development. The introduction explains the primary aim was to investigate how emotional problems, coping flexibility, and age may be related to appraised threat to their psychological needs for relatedness, competence and autonomy and importantly how these relate to their anticipation of their coping strategies. In addition the study aimed to investigate appraised need threat and coping strategy in relation to three domains of stress.  A substantial body of data has been analysed and the results reported in detail. The discussion touches on the implications and possible applications. However, the presentation could be amended to attract greater readership and in turn impact.

***Thank you for these positive comments.

In the abstract, Give the two aims and the unique aspect of this study. State how the results may benefit the target population.  Explain how the findings might be applied in practice, as well as in future research.  

***We have revised the abstract, keeping within the word limit.

In the conclusion, state the aims and emphasise the unique element of this study. Summarise the results, and state how the results may benefit the target population. Explain how the findings might be applied in practice, as well as in future research.  

***We think this recommendation has significantly improved the conclusion. See pp. 27-29:

  1. Conclusions

The purpose of this study was to identify how adolescents’ emotional problems, coping flexibility, and age can shape stress appraisals and ways of coping across three major stressor domains for adolescents – academics interactions with parents, and peer relationships. Incorporating all measures into latent structural equation models, we also tested the associations of stress appraisals with ways of coping. Using videos of stressful events as stimuli, adolescents appraised six stressors, two in each domain, using items that reflected threat to relatedness, competence, and autonomy. Adolescents also reported their anticipated ways of coping with each stressor, which were collapsed into four composite ways of coping: two positive engagement ways of coping labeled as active coping (i.e., support seeking, problem-solving, and information-seeking) and self-reliance, and two negative disengagement ways of coping labeled as withdrawal coping (i.e., social withdrawal, escape, and submission) and helplessness. This study was the first to measure stress appraisal as perceived threat to the psychological needs for relatedness, competence, and autonomy, separately measuring appraisals of portrayals of academic, parent, and peer stressors. It was also the first to model how adolescents’ emotional problems (combined depressive and anxiety symptoms), coping flexibility, and age relate to stress appraisal and intended engagement and disengagement ways of coping both across and within the three stressor domains.

Stress appraisal, when measured as the perception of threat to relatedness, competence, and autonomy, did relate to the ways that adolescents cope with stress. At the general level considering all stressor domains, adolescents who appraised more threat reported they would use more withdrawal to cope and would feel more helpless, but they also reported they would use more active coping and want to be more self-reliant. When domain of stress was considered, some of the associations were stronger within some domains relative to others – two associations stood out as different, with a positive effect of stress appraisal on self-reliance only when the stressor was in the parent-adolescent relationship domain, and the positive effect of stress appraisal on helplessness only significant when the stressor was in the peer relationship domain. Adding to these associations, adolescents’ emotional problems, coping flexibility (belief in the capacity to use multiple ways to cope with stress when needed), and age had influence, with emotional problems having the most far-reaching effects on stress appraisal (more appraisal of threat) and less engagement and more disengagement ways of coping (less active and more problem coping). However, coping flexibility related to more use of engagement ways of coping, and age was uniquely related to more self-reliance and helplessness.

Taken together, the present study findings suggest that interventions for adolescents, which are closely timed to the experience of stressors, could be useful for growth by supporting youth to consider how appraisals can interfere with relatedness, competence, and autonomy (e.g., feeling rejected or excluded, incompetent, or coerced), to consider alternative appraisals, and to understanding how appraisals relate to the many available ways available to cope with stress in the short and in the longer term. This support would acknowledge how coping actions can be relied upon to solve problems and regulate emotion, while also helping adolescents to reflect on their own beliefs and behaviors when facing stressful events [e.g., for more details see 3,72,73]. At the same time, the findings suggest that adolescents could benefit from discussions and education regarding coping with stressors to support their development of coping flexibility. This could be supported by providing ongoing opportunities to practice and receive direct feedback on stress appraisals and ways of coping from parents and teachers or other professionals when hassles, challenges or other stressors occur at home or at school.

Minor suggestion:

Line 418, the word using might be replaced by useful: "....would be particularly useful for interventions with...."

***We have corrected this sentence. Thank you.

Reviewer 3 Report

Comments and Suggestions for Authors

This study aimed to examine whether adolescents’ emotional problems, coping flexibility, and age were related to their appraised threat to the psychological needs for relatedness, competence, and autonomy and how these threats then relate to their anticipated use of engagement and disengagement ways of coping. This study made the new contribution to adolescents’ mental health. I have some questions for this study.

1.      It is hard to follow the authors’ thoughts sometimes because of many issues examined at the same time. For example, the title “Psychological Need Threat and Coping Across and Within Academic, Parent, and Peer Stressors…” The readers may be confused by the terms “need,” “threat,” and ”coping,” as well as “across” and “within” three stressors. Making these terms being easily understood is helpful.

2.      This study examined the associations among the variables by several SEM models. However, only one figure was shown in the manuscript. I would like to suggest the authors to describe more by adding figures as Supplementary materials.

3.      This study examined the roles of emotional problems, coping flexibility, and age in their appraised threat to the psychological needs for relatedness, competence, and autonomy “across” and “within” academic, parent, and peer stressors. It is not clear why both “across” and “within” these three stressors were examined. The reasons should be described.

4.      The differences in the results of examining the roles of perceived threats to psychological need and coping “across” and “within” these three stressors should be emphasized in Discussion. Potential explanations for the differences can be proposed.

5.      Most of the data came from a follow-up study. However, “coping flexibility” seemed to be from the T1 survey. the potential influence should be discussed.

6.      Section 2.3.3. “As part of the T1 survey…” was hard to understand.

7.      Participants: the method of recruiting participants described in 2.2. Procedure should be moved forward to 2.1. Participants.

8.      This study examined participants’ appraised threat of psychological needs by asking them three questions. These three questions were derived from the situations shown in video excerpts. However, there were two excerpts for one kind of stressor. How did the authors select the situation for each stressor?

Comments on the Quality of English Language

Fine

Author Response

Reviewer 3

This study aimed to examine whether adolescents’ emotional problems, coping flexibility, and age were related to their appraised threat to the psychological needs for relatedness, competence, and autonomy and how these threats then relate to their anticipated use of engagement and disengagement ways of coping. This study made the new contribution to adolescents’ mental health. I have some questions for this study.

  1. It is hard to follow the authors’ thoughts sometimes because of many issues examined at the same time. For example, the title “Psychological Need Threat and Coping Across and Within Academic, Parent, and Peer Stressors…” The readers may be confused by the terms “need,” “threat,” and ”coping,” as well as “across” and “within” three stressors. Making these terms being easily understood is helpful.\

***You are right that our terminology needed to be more consistent and clear, and our writing could have been more concise and precise. We have paid particular attention to the issues raised here throughout the paper and hope the revised paper is clearer.

  1. This study examined the associations among the variables by several SEM models. However, only one figure was shown in the manuscript. I would like to suggest the authors to describe more by adding figures as Supplementary materials.

***We had attempted to construct a figure to illustrate all of the paths tested in Model 2, but we had determined it was too unwieldy to display all of the paths this way. So, overall, we didn’t think this was helpful to include. Instead, we have included Table 3 that lists all directional paths in this model.

  1. This study examined the roles of emotional problems, coping flexibility, and age in their appraised threat to the psychological needs for relatedness, competence, and autonomy “across” and “within” academic, parent, and peer stressors. It is not clear why both “across” and “within” these three stressors were examined. The reasons should be described.

***We have modified sentences throughout the Introduction in attempt to clarify our purpose and why we felt it was important to consider general/across domain findings and to estimate associations within each stressor domain (academic, parent, peer).

  1. The differences in the results of examining the roles of perceived threats to psychological need and coping “across” and “within” these three stressors should be emphasized in Discussion. Potential explanations for the differences can be proposed.

***We have added a little more detail regarding these differences and now include a figure that shows the paths.

  1. Most of the data came from a follow-up study. However, “coping flexibility” seemed to be from the T1 survey. the potential influence should be discussed.

***Both emotional problems and coping flexibility were measured about 6 months earlier during the T1 survey.  We added this to the limitations section on p. 27:

Moreover, both emotional problems and coping flexibility were measured about 6 months prior to the measurement of stress appraisal and coping. It is possibility this introduced some error, given that emotional problems and flexibility could have changed over the six months. It is not clear how this could have influenced the findings reported here, but it is most likely this would have reduced the size of associations rather than inflating them.

  1. Section 2.3.3. “As part of the T1 survey…” was hard to understand.

***We have modified this sentence to read, As part of the T1 survey completed about 6 months prior to this study, participants were prompted with a definition of stress and coping before completing 6 items from the Self-perceived Flexible Coping Scale (SFCS; e.g., “I can come up with lots of ways to make myself feel better if I am stressed”) [46].

  1. Participants: the method of recruiting participants described in 2.2. Procedure should be moved forward to 2.1. Participants.

***We have merged and reorganized the Participants and Procedure section. see pp. 8-9:

2.1 Participants and Procedure

Participants ranged from 10 to 15 years of age (M = 12.5, SD = 1.5), with 195 boys (48%), 208 girls (50%), and 7 (2%) nonbinary/other (total N = 410). Adolescents could select multiple of the following to describe themselves: White; Asian; Australian First Peoples, Torres Strait Islander or Pacific Islander; other race/ethnicity; born in Australia; and born in New Zealand. Overall, 99% of adolescents ticked at least one answer, with 65% reporting White, 9% Asian, 5% Australian First Peoples, Torres Strait Islander or Pacific Islander, 37% other race/ethnicity, 85% born in Australia, and 7% born in New Zealand. Students attended grades 7 to 10 at three Australian secondary schools (ages 13-15, 56%) or were in grades 5 or 6 at five of their feeder schools (ages 10-12, 44%). School websites reported that 14%–29% of students fell into the lowest and 4%–30% into the highest income quartiles, whereas the proportion of students who spoke a language other than English at home ranged from 5%–29%.

The participants in this study were drawn from a longitudinal study six months after the first wave of data collection (T1) had been completed. Students were selected to be contacted about participation in the present study from among 677 parents (78%) who had agreed to be contacted for other research studies. The aim was to achieve a random 50% participation rate in the present study, expecting this to provide a good representation of the larger sample. The randomly selected students were contacted via phone or email by research assistants to gain additional consent for participation. Parents and students were introduced to the study protocol and provided with a link to an online survey with embedded film clips. Once assenting to participate, adolescents were recontacted up to seven times via phone or email to remind them about survey participation. Data were collected in two periods: July-Oct 2020 and Mar-June 2021 to avoid data collection at the end of the school year and in the Australian summer break. The datasets generated during and/or analysed during the current study are available from the corresponding author on reasonable request.

  1. This study examined participants’ appraised threat of psychological needs by asking them three questions. These three questions were derived from the situations shown in video excerpts. However, there were two excerpts for one kind of stressor. How did the authors select the situation for each stressor?

***We have added more content to the information about the film clips to summarize the extensive preliminary work and pilot testing that has been done over the years to develop this analog procedure. see pp. 9-10:

2.2.1. Video Excerpts

Adolescents viewed six short (< 30 second) film clips of stressors involving academics/school (2 clips), parents (2 clips), and peers (2 clips). The stressful scenes depicted were as follows: A girl suspected of cheating by a teacher at school, a girl finding out that she did much worse than she expected on a school written assignment, a boy having a verbal argument with his father, a girl witnessing a loud argument between her parents, a boy who was the last one picked for a team, and a boy being teased and laughed at by classmates. A positive scene of a group at a birthday party was shown mid-point to distract from the patterns of stressors. All stressors are commonly experienced and have been found to be distressing for adolescents [44,55,56]. The film clips were approximately 30 seconds in length and depicted an adolescent appearing close in age to the participants, as the central figure. All scenes were in English and were from General (G) rated films or YouTube. This analogue procedure was developed in multiple previous studies that involved numerous pilot tests of written vignettes and film clips, then refining the vignettes and film clips to include daily hassles (rather than major stressful life events, e.g., loss of a parent) and events that were reported to be commonly experienced (33,34,57-62).

Reviewer 4 Report

Comments and Suggestions for Authors

Dear authors, please find attached the revision of your article

Author Response

Reviewer 4

The summary is not very clear. By reading the objectives, the reader gets the impression

that one of the independent variables is adaptation and that the dependent variable is

adaptation. Could the authors please clarify? Furthermore, it appears that the

independent variables used are not of the same order. Some are invoked and others not. Generally speaking, the reader gets lost in the experimental plan which is not presented in a canonical manner in this summary. The variables are very numerous and not always

titled in the same way. Authors should provide a general hypothesis that is justified by

existing theoretical controversy and provide less detail about the protocol there.

***You are right that our writing could have been more concise and precise. We have paid particular attention to the issues you have raised while revising the paper and hope the paper is clearer.

The entire section from line 50 to line 60 is not documented. Authors should justify the

ideas stated on the basis of existing preliminary work.

***We were not sure which lines were not documented.

The objective of the study is not very clear because the authors use a very varied lexicon

to describe the same process. The notion of quality of life, for example, appears without

being clearly defined.

***We cannot find the term "quality of life" in the paper.

Why was ethnic origin asked? Is this a variable in the experimental design? If so, what

preliminary work is this variable based on? Based on the General Data Protection

Regulation, the collection of information on ethnic origin can reinforce

communitarianism and identity-based behavior if it is not necessary for scientific

research purposes. Could the authors justify this? The same applies to the type of school

attended. Why did you ask this question? Is this a new independent variable? In

summary, demographic data seems far from the point. However, more precise

information on the psychological characteristics of students (such as being followed by

a psychiatrist or a doctor for other health problems) does not appear among the

inclusion or exclusion criteria.

***It is standard practice to gather this type of information if only for the purpose of describing the participants to help the reader to consider the generalizability of the findings.

Concerning the material, have the video extracts been pre-tested beforehand as

presenting stress factors? Even though the situations are considered stressful by

research, many variables involved in videos can generate mixed emotions. It is essential

to pre-test the material with a sample of adolescents (girls and boys) in the chosen age

group in order to ensure that these situations concern them directly and that each extract

has the desired effect. Likewise, it is not easy for a boy to “project himself” into a scene

where he sees “a girl” who is having an altercation, for example. Were the videos

gendered and presented according to gender to the adolescent concerned? (same for

ethnicity).

***We completely agree. We have added more content to the information about the film clips to summarize the extensive preliminary work and pilot testing that has been done over the years to develop this analog procedure. see pp. 9-10:

2.2.1. Video Excerpts

Adolescents viewed six short (< 30 second) film clips of stressors involving academics/school (2 clips), parents (2 clips), and peers (2 clips). The stressful scenes depicted were as follows: A girl suspected of cheating by a teacher at school, a girl finding out that she did much worse than she expected on a school written assignment, a boy having a verbal argument with his father, a girl witnessing a loud argument between her parents, a boy who was the last one picked for a team, and a boy being teased and laughed at by classmates. A positive scene of a group at a birthday party was shown mid-point to distract from the patterns of stressors. All stressors are commonly experienced and have been found to be distressing for adolescents [44,55,56]. The film clips were approximately 30 seconds in length and depicted an adolescent appearing close in age to the participants, as the central figure. All scenes were in English and were from General (G) rated films or YouTube. This analogue procedure was developed in multiple previous studies that involved numerous pilot tests of written vignettes and film clips, then refining the vignettes and film clips to include daily hassles (rather than major stressful life events, e.g., loss of a parent) and events that were reported to be commonly experienced (33,34,57-62).

In addition, we added a limitation regarding the gender of the central character in each video clip. see p. 27:

Third, all participants viewed the same film clips and the clips were not randomized. Therefore, boys viewed some clips with a girl as the central character and vice versa. This was done to maintain standardization, but this could have affected some of the data collected or the findings if this interfered with immersion in the content or there were order effects that were undetected.

Questioning anxiety and depression seems to disconnect from the subject's history. This

refers to the criticism made above. Indeed, some adolescents may present with an

anxiety disorder and/or depression and/or school phobia, but some may have already

been diagnosed and can already be monitored for these difficulties. Did the authors

make sure of this?

***We did not ask about diagnosis so have been careful to acknowledge and refer to our measure as capturing level of emotional problems only.

Regarding the results, table 1 is not very readable. It would be important to remember

in the table what the numbers 1, 2, 3 correspond to… In addition, this table extends over

2 pages. Would a synthesis be possible?

***We have separated the content into two tables (Tables 1 and 2), hoping that this will be easier to read.

Round 2

Reviewer 4 Report

Comments and Suggestions for Authors

Unfortunately, the authors did not address the majority of the major points for revision. Important methodological and ethical clarifications need to be made before this manuscript can be published in the target journal.

Author Response

Reviewer Comments: Unfortunately, the authors did not address the majority of the major points for revision. Important methodological and ethical clarifications need to be made before this manuscript can be published in the target journal.

Response: We are uncertain how to respond to this very general comment. In the previous review, the Reviewers had asked for clarification of the study design and objectives, reasoning for gathering descriptive information about the participants, asked for more details about the film clips, raised the issue of diagnoses of depression or other emotional problems in the participants, and asked for some changes to Table 1 to make it easier to read.  We have nothing to add to out responses to these queries and suggestions here.

We did check the manuscript regarding our description of the Human Research Ethics guidelines we followed, our approval by our University;s Human Ethics Review Committee prior to conducting this study, and our description of gathering parent consent and student assent to participate. See the highlighted text on p. 5:

All aspects of this study were approved by the Griffith University Human Research Ethics Committee (protocol # 2019-178) prior to contacting schools and parents. The study was carried out in accordance with the World Medical Association Declaration of Helsinki. 

Round 3

Reviewer 4 Report

Comments and Suggestions for Authors

Thank you